# The UHPLC-Orbitrap MS/MS and Network Pharmacology Strategies Reveal the Active Antioxidants of Bleeding Sap from Sponge Gourd in Treating Tuberculosis

**DOI:** 10.3390/ijms262010231

**Published:** 2025-10-21

**Authors:** Di Zhang, Lu Jiang, Yujiang Dai, Xinxin Si, Huifang Li, Komal Anjum

**Affiliations:** 1Jiangsu Institute of Marine Resources Development, Jiangsu Ocean University, Lianyungang 222005, China; dizhang@jou.edu.cn (D.Z.); jianglu0802@163.com (L.J.); 15736591844@163.com (Y.D.); sixx@jou.edu.cn (X.S.); 2Department of Medicine and Pharmacy, Ocean University of China, Qingdao 266000, China

**Keywords:** *Luffa cylindrica*, bleeding sap, tuberculosis, antioxidant, network pharmacology

## Abstract

The bleeding sap of *Luffa cylindrica* (L.) Roem has been used for the treatment of tuberculosis since the record of *Supplements to Compendium of Materia Medica*. The active components and possible mechanism of it are yet ambiguous. Hence, this study is focused on investigating the possible mechanism underpinning this effect on the perspective of the antioxidant ingredients from the bleeding sap. Through organic solvents extraction, HPLC fractionation, DPPH trials evaluation, and UHPLC-Orbitrap tandem MS identification, a total of 37 compounds were identified from the bleeding sap with the strongest antioxidant ability. Network pharmacology, bioinformatics, and molecular docking as well as literature review revealed 13 compounds, including linoleic acid, abietic acid, and tretinoin, that might exert their anti-tuberculosis function via actions with PPARγ or MAPK pathway. These findings offer guidance for the potential applications of *Luffa cylindrica* (L.) Roem as a functional food.

## 1. Introduction

*Luffa cylindrica* (L.) Roem, often known as sponge gourd or loofah, belongs to the *Cucurbitaceae* family. In China, as a food or herbal drug, *Luffa cylindrica* (L.) Roem has been used for at least 1600 years [1]. In fact, the whole plant of *Luffa cylindrica* (L.) Roem is an amazing medical treasure. For example, its fruit is able to reduce blood sugar [2,3], protect the liver [4,5,6], and fight inflammation [7,8]; its leaves have been reported to promote anti-cancer [9,10], hepaprotective [11,12] and antioxidant activity [13]; the *retinervus luffae fructus* of it possesses a hypolipidemic activity [14,15] and a diuretic effect [16] and can improve heart function [17,18]; the stems have anti-inflammation, antioxidant, and anti-allergic effects [19,20,21,22]; while the seeds have been demonstrated to promote anti-cancer and anti-HIV activity [23,24,25]. The diversity of pharmacological activity of *Luffa cylindrica* (L.) Roem depends on the complication of its chemical constituents. The phytochemical investigation of *Luffa cylindrica* (L.) Roem reveals that it mainly contains triterpenoids, flavonoids, glycosides, proteins, and tannins [26,27,28,29,30].

The bleeding sap collected from the fresh and robust wine of sponge gourd, known as “tian luo water” in Chinese traditional medicine, has also been used for the treatment of tuberculosis since the Qing dynasty [31]. As is well known, tuberculosis (TB) is an infectious respiratory disease caused by *Mycobacterium tuberculosis* (*M. tb*), causing symptoms such as persistent cough, sputum production, and hemoptysis [32]. However, studies until now have shown only that the bleeding sap is capable of antioxidant and anti-fungal activity [21,22,33] but have not demonstrated how the bleeding sap cures tuberculosis. Neither its mechanism nor the chemical foundation back up remains ambiguous [22].

There is a relationship between the antioxidant activity of sponge gourd bleeding sap and its ability to treat tuberculosis. In this study, we firstly screened the antioxidant ability of the bleeding sap by combining different organic solvents extraction and preparative High Performance Liquid Chromatography (pre-HPLC) subfraction as well as 2,2-diphenyl-1-picrylhydrazyl (DPPH) assay [34]. Then, the ultra-high-performance liquid chromatography-tandem mass spectrometry (UHPLC-MS/MS) was applied for clarifying its components [35], followed by the network pharmacology and GEO database [36,37] to analyze the relationships between sponge gourd bleeding sap and its treatment of tuberculosis. At last, the molecular docking [38,39] was utilized for verifying this relationship.

## 2. Results

### 2.1. The Group with the Highest Antioxidant Capacity

The DPPH free radical scavenging assay demonstrated that different parts of bleeding sap of sponge gourd possessed different antioxidant capacity (Appendix A). As shown in Figure 1A–D, the non-linear increasing antioxidant capacities of P1, 2B, 2D, and 2E were consistent with their concentration growth, especially when they were at the same concentration of 0.4 mg/mL (Figure 1A). The scavenging rates of 2B, 2D, and 2E were notably higher than that of the other extractions. In order to explore which part was in possession of the strongest antioxidant ability, bioassays of a series of concentration gradient experiments were performed, which demonstrated that the extraction 2E had the best antioxidant capacities (Figure 1B,C). Meanwhile, it was found that its scavenging rate was highest when its concentration was 2.5 mg/mL (Figure 1D).

### 2.2. Compounds Identification

The UHPLC-MS/MS experiment analysis elucidated what sorts of components resulted in the significant antioxidant ability of the bleeding sap fraction 2E. The total ion chromatograms (TICs) in positive and negative modes are shown in Figure 2A,B. A total of 37 compounds, including 16 fatty acids (**1**–**6**, **9**–**18**), 2 fatty acid esters (**7**, **8**), 4 phosphides (**19**–**22**), 4 alkaloids (**23**–**26**), 3 sulfides (**27**–**29**), 3 aromatics (**30**–**32**), 2 terpenes (**33**, **34**), 1 flavonoid (**35**), 1 ether (**36**), and 1 steroid (**37**), were identified via the mzCloud database according to their match degree given by Compound Discoverer^TM^ 3.0 software (Appendix A). The detailed information, such as the mzCloud best match degree, the formula, name, and the annotation delta mass, are summarized in Table 1.

#### 2.2.1. Identification of Fatty Acids and Esters

The identified fatty acids can be further divided into four categories, including eight saturated fatty acids (**1**, **3**–**6**, **9**, **10**, **14**), two hydroxyl saturated ones (**2**, **18**), four unsaturated ones (**11**–**13**, **15**), as well as two hydroxyl unsaturated ones (**16**, **17**). Their fragment ions can be found in Appendix A. For all fatty acids, there is a strong [M-H]^−1^ peak on the negative mode of MS^1^ spectra and a common [M-H-H_2_O]^−1^ peak with weak intensity on the MS^2^ spectra, respectively. Moreover, the fragment ion abundance was inclined to an enhancement with the existence of hydroxy group or more double bonds [40]. As for two fatty acid esters (**7** and **8**), more abundance and complicated ions were observed from the positive spectra of **7** than that of **8** on the negative mode. The possible fragment pathways of **7** are shown in Figure 3A (Appendix A).

#### 2.2.2. Identification of Phosphorus-Containing Compounds

All the organophosphorus compounds (Appendix A) showed the common [M+H]^+1^ or [M+Na]^+1^ peak on their MS^1^ spectra at the positive mode, and three of them (**19**, **20**, **22**) possessed characterized MS^2^ fragment ions of [H_4_PO_4_]^+^ at a mass unit of 98.9840. Moreover, due to the exististence of benzene-substitute, the fragment behaviors of compound **21** were more complicated than that of the others. The typical fragment pathways of **22** are shown in Figure 3B (Appendix A).

#### 2.2.3. Identification of Alkaloids

The only rule for four alkaloids (**23**–**26,**
Appendix A) observed from their positive MS^1^ spectra was that they all shared a strong [M+H]^+1^ peak, respectively. As a consequence of different types of alkaloids, their fragment behavior of MS^2^ were greatly distinguishable. For the amide alkaloid **23**, its MS^2^ fragment ions were obtained by degradation of methylene groups from [M+H]^+1^ peak. Except for the [M+H]^+1^ peak, the [M+Na]^+1^ and [M+K]^+1^, together with [2M+Na]^+1^ peak of pyrimidine **24**, were obtained from its MS^1^ spectrum in positive mode (Appendix A). Furthermore, its MS/MS spectrum yielded a limited intensity peak of [M+H-H_2_O]^+1^ at 107.0603 Da. The MS^2^ spectra of quinoline (**25**) displayed more product ions, of which the possible fragment pathways are shown in Figure 3C (Appendix A).

#### 2.2.4. Identification of Sulfur-Containing Compounds

Three sulfides (**27**–**29**, Appendix A) shared a strong intensity [M-H]^−1^ peak at their negative mode of MS^1^ spectra, respectively; in addition, their MS^2^ spectra possessed the same daughter ions of [SO_3_]^−^ at a mass unit of 79.9574. The peak of 96.9601 Da with strong intensity appeared in the MS^2^ spectra of **27** and **28**; it is proposed to be the ion of [HSO_4_]^−^. Moreover, there is also a strong peak at 183.0119 Da in the MS^2^ spectra of **29**, which was presumed to be the [C_8_H_7_O_3_S]^−^.

#### 2.2.5. Identification of Aromatics

The [M-H]^−1^ peaks with strong intensity of two phenols, **30** and **32**, were observed from their MS^1^ spectra in the negative mode, respectively. Furthermore, there is a strong peak of 163.1127 Da displayed in the MS^2^ of **32**, which was inferred to be [C_11_H_15_O]^−^, followed by a peak at 147.0812 Da caused by degradation of a neutral CH_4_ fragment. The same pathway of degrading the neutral CH_4_ could be observed from 205.1595 Da ([M-H]^−1^) to 189.1284 Da ([C_13_H_17_O]^−^) in the MS^2^ spectrum of **30**. As for **31**, an [M+H]^+1^ ion with strong intensity was obtained from its positive MS^1^ spectrum, and a typical [C_8_H_5_O_3_]^+^ ion of 149.0232 Da yielded from its MS^2^ spectrum. Their fragment ions of all aromatic’s compounds can be found in Appendix A.

#### 2.2.6. Identification of Other Compounds

The [M-H]^−1^ ions of two terpenes, **33** and **34**, were obtained from their MS^1^ spectra, respectively; moreover, on their MS^2^ spectra, the [M-H-COO]^−1^ ions were observed at 257.1539 and 255.2118 Da. For the flavonoid (**35**), ether (**36**) and steroid (**37**), their [M+H]^+1^ peaks of strong intensity were gained from their MS^1^ spectra in the positive mode. Additionally, the probable fragmentation pathways of their MS/MS ions are shown in Figure 3D–F (Appendix A).

### 2.3. Network Pharmacology Analysis

#### 2.3.1. Intersecting Targets Between the Protein Targets of Identified-Compounds and TB-Related Genes

Through screening of the 37 compounds in the SwissTargetPrediction database, a total of 323 potential and non-overlapping protein targets were identified. Meanwhile, with “tuberculosis” as a keyword, an aggregation of non-repetitive 1695 TB-related targets were obtained through searching the OMIM, GENECARDS, and DISGENET databases. Intersecting targets between the protein targets of compounds and TB-related genes were then achieved by using Venny 2.1.0 webpage. Finally, an integration of 99 intersecting targets were obtained, as shown in Figure 4A.

#### 2.3.2. The PPI Network Construction and Screening of Core Targets

A PPI network which involved 99 nodes and 921 edges, as shown in Figure 4B, were then constructed by subjecting 99 intersecting targets to the STRING database. The PPI network was further analyzed with the centiscape 2.2 algorithm through the filtering thresholds for “closeness unDir > 0.005”, “betweenness unDir > 96.08”, and “degree unDir > 18.60”, as well as by the topological analysis of CytoNCA, to yield 23 potential anti-tuberculosis core targets, which contain 23 nodes and 182 edges, as visualized in Figure 4C.

#### 2.3.3. Analysis of GO Enrichment

The 23 core targets were entered into the DAVID database to fulfill the GO enrichment analysis. With a threshold of *p* < 0.05 set as the screening parameters, the top 10 entries with noticeable enrichment of biological processes (BPs), cellular components (CCs), and molecular functions (MFs) are, respectively, presented in Figure 4D. For BPs, the enrichment of three biological processes, including negative regulation of miRNA transcription, positive regulation of ERK1 and ERK2 cascade, and response to xenobiotic stimulus, seemed more reliable than that of other processes; CCs involved the endoplasmic reticulum membrane, cytoplasm, and caveola, and the endoplasmic reticulum lumen was enriched; and for MFs, the core targets were significantly enriched in enzyme binding, aromatase activity, steroid hydroxylase activity, monooxygenase activity, heme binding, oxidoreductase activity, oxygen binding, iron ion binding, and so on. Furthermore, this sort of enrichment was sequential, that is, of MFs > BPs > CCs.

#### 2.3.4. Analysis of KEGG Pathways Enrichment

The same condition was applied for the KEGG pathways enrichment analysis. The top 20 pathways, including the lipid and atherosclerosis pathway, the AGE–RAGE signaling pathway in diabetic complications, human cytomegalovirus infection, the IL-17 signaling pathway, and so on, were obtained from the KEGG analysis (Figure 4E). According to the *p* value, the lipid and atherosclerosis pathway was the most related to the core targets, meaning that the targets on this pathway might be the therapeutic targets for the antioxidant components of bleeding sap in the treatment of tuberculosis.

### 2.4. Bioinformatics Analysis

A total of 10,772 targets were obtained by screening the sample GSE189996. The clinical targets were then imported into the Venny 2.1.0 webpage together with 23 core ones to yield 18 intersecting points (Figure 5A). To make a better understanding of the biological processes and pathways of the 18 intersecting targets, we performed GO and KEGG pathway analyses using the DAVID database (Figure 5B,C), which showed that the lipid and atherosclerosis signaling pathway was the one with the most significant differences. By integrating this result with the one in 2.3.4 mentioned above, it was revealed that the targets in the lipid and atherosclerosis, including IL6, CXCL8, CASP3, MAPK1, PRKCA, PPARγ, MAPK14, TNF, TP53, ICAM1, MAPK3, could be the clinical therapeutic targets. Finally, the compound-target-pathway (CTP) network was constructed using Cytoscape 3.9.0 software, as shown in Figure 5D, which consisted of 13 compounds, 11 targets, and 1 pathway.

### 2.5. Molecular Docking

To investigate the interaction between the active compounds and the targeted proteins, the molecular docking via AutoDock Vina 1.2.0 software was applied for evaluating the binding affinity between them. When the docking energy was less than −5 kcal/mol, the interaction between the molecule and protein was recognized as reliable, further supporting that the molecule is a potential ligand for the receptor protein [73]. The results showed that the main compounds were able to tightly bind with their targets, of which the biggest binding energy was −5.02 kcal/mol and the smallest one was −8.21 kcal/mol, as shown in Table 2. Looking at the targeted protein, the PPARγ (peroxisome proliferator-activated receptor γ) was the most extensive receptor, which was able to bind with seven different ligands, including five fatty acids—stearic acid (**9**), oleic acid (**11**), trans-10-heptadecenoic acid (**12**), palmitoleic acid (**13**), as well as linoleic acid (**15**)—and two terpenes, including abietic acid (**33**) and tretinoin (**34**), with their affinity energy ranging from −8.21 to −5.04 kcal/mol. Moreover, both targeted proteins, MAPK3 (mitogen-activated protein kinase 3) and MAPK14 (mitogen-activated protein kinase 14), showed a better affinity with two or three different compounds, respectively. On the other, the abietic acid exhibited its affinitive abilities with several distinct targeted proteins, which contained the TNF (tumor necrosis factor), TP53 (tumor protein 53), PPKCA (protein kinase c alpha), MAPK3, and PPARγ, with the lowest binding energy being −8.21 kcal/mol while the highest one was −6.06 kcal/mol. Additionally, the compounds dibutyl phthalate (**31**) and tretinoin (**34**) could have also interacted with several targeted proteins with disparate hydrogen bonds. The representative molecular dockings are listed in Figure 6A–J.

## 3. Discussion

Tuberculosis, a chronic lung inflammatory disease produced by *M. tb* infection, is still a major public risk with a high human mortality rate [74,75]. The currently therapy for tuberculosis mainly involves the application of antibiotics, which leads to the increasing existence of multi-drug tolerance, subsequently triggering another public health concern [76]. In this case, using traditional herbal medicine for treating TB has become another valuable and prospective approach. As a kind of traditional herbal medicine, the bleeding sap from *Luffa cylindrica* (L.) Roem has been used for the treatment of TB for almost 260 years [77]. In this study, we have attempted to elucidate the mechanism of its curing effect with TB using a combinational application of in vitro antioxidant experiment, HPLC fraction, UPLC-Orbitrap tandem MS and network pharmacology as well as molecular docking.

Notably, it is the first time that the UPLC-Orbitrap tandem MS was applied for the identification of the chemical components of bleeding sap. The phytochemical investigation of the strongest antioxidant parts revealed that the bleeding sap contained at least ten sorts of different chemical ingredients, among which the most abundant one was the saturated or unsaturated long-chained fatty acids, leading to an acidic pH of bleeding sap [78]. Interestingly, four discrete organophosphate compounds were identified from the bleeding sap for the first time, although they probably were absorbed from the soil, indicating the possible toxicity of bleeding sap considering that bleeding sap has been used for skin protection for a long time, too. Meanwhile, the untargeted identification also revealed three previously unreported sulfates from the bleeding sap. Morover, the flavone, terpene, ether, and steroid were also reported from bleeding sap for the first time.

With regard to the possible mechanism of how these antioxidant agents treat TB, the network pharmacology coupled with bioinformatics analysis and molecular docking may have provided some clues. By taking the intersection between the core targets predicted from the antioxidant agents and the clinical targets of GSE189996, the most probable involved therapy was determined to be the lipid and atherosclerosis signaling pathway, upon which the compound-target-pathway network of sponge gourd bleeding sap was constructed for the first time. Based on the network and molecular docking, the PPARγ was the most possible therapeutic target.

It has been reported that the PPARγ can exert its anti-inflammatory action not only through inhibiting the gene expression of nuclear factor NF-κB (nuclear factor kappa-light-chain-enhancer of activated b cells) [79] but also through inhibiting the expression of IL-1α/2/6/12 (interleukin 1α/2/6/12) and TNF-α (tumor necrosis factor α) as well as TGF-β (transforming growth factor-β) [80]. Meanwhile, it also acts as a key transcriptional regulator involved in lipid metabolism and storage, especially in adipogenesis and lipid synthesis [81,82].

The antioxidant agent, for example, abietic acid (**33**), has been reported to possess an anti-inflammatory effect due to the activation of PPARγ in macrophages [83], while the tretinoin (**34**) can also regulate the neural crest cell fate by differentiating into adipocyte cells through the PPARγ pathway, and as a natural ligand of PPARγ, linoleic acid (**15**) can affect the fatty acids’ transportation, whatever they are, from endogenous de novo synthesis or exogenous [50]. Although the *M. tb* in infected macrophages are able to increase the expression of PPARγ and lower the latter’s ability to exterminate the infections [84,85], the activated PPARγ still can relieve the progress of TB and repair the pathological damage to lung tissue, through ameliorating *M. tb*-induced foamy macrophage formation via the ABCG1 (ATP-binding cassette transporter G1)-dependent cholesterol efflux [86] or via the lipid droplets secreting from the antimicrobial peptide at some time [87], as well as through improving innate immune response through the transformation and proliferation of T lymphocytes [88].

Regarding the “positive regulation of ERK1/2 (extracellular signal regulated kinase 1/2) cascade” in the biological process analysis within GO enrichment statics, the antioxidants might also act with the MAPK pathway to inhibit the growth of mycobacteria. Studies have revealed that the activation of the MAPK pathway can lead to the expression of inflammatory cytokines, such as TNF-α, IL-10, and MCP-1 (monocyte chemoattractant protein-1), to aggravate TB [89,90], and the suppression of it may attenuate TB progression [91,92]. The tretinoin (**34**) or retinoic acid is able to improve therapeutic efficacy when combined with antibiotic application for treating tuberculosis through regulating MEK/ERK (methyl ethyl ketone/extracellular regulated protein kinases) and p38 MAPK pathway [92]. As for abietic acid (**33**), although there is not yet a report on its anti-tuberculosis effects through interactions with the MAPK pathway, studies about its ability to attenuate osteolysis and pancreatic injury through this pathway have been reported [93,94]. Moreover, the pervasive environmental fomite, dibutyl phthalate (**31**), might affect the progress of TB through elevating the expression of caspase 3 to induce apoptosis or change the oxidative stress to produce reactive oxygen species [95,96], although it also can induce cell proliferation or tissue damage by activating the MAPK pathway [97,98].

## 4. Materials and Methods

### 4.1. Materials and Instruments

The sponge gourd bleeding sap was purchased from the local farmers in Lianyungang, China. Solvents of HPLC grade such as methanol (MeOH) and acetonitrile (MeCN) were all brought from Aladdin Industrial Co., Ltd., Shanghai, China. Vitamin C, 2,2-diphenyl-1-picrylhydrazyl and solvents of analysis grade such as ethyl acetate (EA), MeOH, acetic acid, and n-butanol were ordered from Shanghai Lingfeng Co., Ltd., Shanghai, China. Moreover, 96-well plates were ordered from Zhoushan Sheng Hong Co., Ltd., Zhejiang, China. A BioTek Synergy Neo2 Hybrid Multimode Reader (Agilent Technologies Inc., California, USA) was used for detecting samples’ absorbance under the wavelength of 517 nm. Pre-HPLC subfraction was performed on the CXTH LC-3000 system equipped with a CT-30 Fuji-C_18_ column (280 × 30 mm, 10 μm) and a UV/Vis detector (Chuang Xin Tong Heng Co., Ltd., Beijing, China). UHPLC-MS/MS was performed on a Thermo Scientific Orbitrap Exploris 120 Mass Spectrometer equipped with Vanquish UHPLC and HESI-II ion source (Thermo Fisher Scientific Co., Ltd., Waltham, MA, USA).

### 4.2. Organic Solvents Extraction with the Bleeding Sap

First, 30 L of bleeding sap was extracted with the same volume of EA three times; then the EA extraction was combined and the solvents were removed in vacuo to obtain part 1 (P1, 143.5 mg). The remaining bleeding sap was then treated with n-butanol in the same way to obtain part 2 (P2, 16.2 g). The last remaining bleeding sap was concentrated to give the dry solids, part 3 (P3, 7.5 g) (Figure 7).

### 4.3. Pre-HPLC Subfraction

The n-butanol extraction P2, was further separated by the preparative CXTH LC-3000 system eluted with MeOH/Acid-H_2_O (1 mL acetic acid dissolved into 1 L H_2_O, *v*/*v*) in gradient (10–100% MeOH) at the UV wavelength of 254 nm and flow rate of 10 mL/min to yield five subfractions: 2A–2E (2A: t*_R_* 5.00–10.00 min, 710.0 mg; 2B: t*_R_* 10.01–15.00 min, 21.1 mg; 2C: t*_R_* 20.01–25.00 min, 15.0 mg; 2D: t*_R_* 25.01–35.00 min, 17.0 mg; 2E: t*_R_* 35.01–50 min, 15.2 mg) (Figure 7).

### 4.4. The DPPH Assay

The antioxidant capacity of different parts of bleeding sap was evaluated by DPPH free radical scavenging assay. In the darkness, the DPPH was dissolved into MeOH to prepare for 0.1 mmol/L solution. All samples, including parts 1–3 and 2A–2E, were prepared for different concentrations in MeOH, respectively. Mix 100 μL of samples with the same volume of 0.1 mmol/L DPPH solutions evenly in 96-well plates, and avoid the light for 30 min. The absorbance was detected by the Synergy Neo2 and was recorded as A1. The absorbance detected by mixing 100 μL of samples with the same volume of MeOH was A2. Under the same conditions, the absorbance of Ad was obtained from the mixture of 100 μL of MeOH and DPPH solutions. Vitamin C was used for the positive control. All experiments were repeated three times. The DPPH radical scavenging rate was as follows: (R) = [1 − (A1 − A2)/Ad] × 100%.

### 4.5. Conditions for UHPLC-MS/MS

Chromatographic conditions: Hypersil GOLD AQ column (100 × 2.1 mm, 1.9 μm, Thermo Fisher Scientific, Waltham, MA, USA); mobile phase A: acetonitrile, mobile phase B: water (0–30 min: 5–50%; 30–35 min: 50–70%; 35–60 min: 70–100%); flow rate: 0.4 mL/min.

Ion Source Parameter Settings: spray voltage: 3.5 kV (+)/3.2 kV (-); capillary temperature: 320 °C; sheath gas: 35 arb; aux gas: 10 arb; sweep gas: 0 arb; probe heater temperature: 350 °C; s-lens S: 60.

Mass Spectrometry Scanning Parameter Settings: scan mode: full MS-ddms^2^; full MS scan range: 100 to 1500 *m*/*z*; spectrum data type: profile; resolution: full MS: 70,000, MS/MS: 17,500; AGC target: full MS: 1 × 10^6^, MS/MS: 2 × 10^5^; maximum IT: full MS: 100 ms, MS/MS: 50 ms; loop count: 3; MSX count: 1; isolation width: 1.5 *m*/*z*; NCE (stepped NCE): 20, 40, 60; minimum AGC target: 8 × 10^3^; intensity threshold: 1.6 × 10^5^; dynamic exclution:5 s.

The results were analyzed through the mzCloud database with Compound Discoverer^TM^ 3.0 software (Thermo Fisher Scientific, Waltham, MA, USA).

### 4.6. Network Pharmacology

#### 4.6.1. Collection of the Active Compounds and Screening Their Potential Targets

The two-dimensional structures of the identified compounds were retrieved from PubChem [99], and imported into the SwissTargetPrediction database [100] for targets prediction. “Prob > 0.1” was used as the screening condition.

#### 4.6.2. Prediction of TB Targets

The OMIM [101], GENECARDS [102], and DISGENET [103] databases were used for predicting the targets of TB (Appendix A). After removing the duplicates among three databases, the final targets integration was obtained (Appendix A).

#### 4.6.3. Protein–Protein Interaction Analysis

The targets predicted by the compounds and the integrated TB targets were entered into the Venny 2.1.0 [104] diagram to obtain the intersecting ones (Appendix A). The intersecting targets were then inputted into the STRING database [73,105] with the Homo Sapiens setting as the selection condition to give the protein–protein interaction (PPI) network. The PPI analysis results were imported into Cytoscape 3.9.0 software, and the core targets were obtained according to the calculation of closeness, betweenness, and degree in centiscape 2.2 algorithm (Appendix A).

#### 4.6.4. KEGG and GO Enrichment Analysis

The core targets were transferred into the DAVID database [106] for identifying core KEGG and GO Pathways (Appendix A).

### 4.7. Bioinformatics Analysis of Tuberculosis

#### 4.7.1. Data Collection

A clinical cohort in the form of GSE189996 data file was retrieved from the GEO database [107], including 58 control cases and 64 tuberculosis cases. The dataset was analyzed with GEO2R (Appendix A) under the parameters of *p* < 0.05 and |Log_2_FC| > 0.

#### 4.7.2. Differentially Expressed Genes and Enrichment Pathways Analysis

The targets screened from clinical dataset GSE189996 and the intersecting targets predicted by the compounds were intersected and visualized by Venny diagram (Appendix A). Meanwhile, KEGG and GO enrichment analyses were also performed using the DAVID database (Appendix A).

#### 4.7.3. Visualization of the Compounds-Targets-Pathways Network

Depending on the KEGG and GO analysis results, the related pathways, core targets, and compounds were corresponded one-by-one and imported into Cytoscape 3.9.0 software to yield the visualized compounds-targets-pathways network.

### 4.8. Molecular Docking

The crystal structures utilized for molecular docking studies include CXCL8 (PDB-ID: 8IC0), TNF (PDB-ID: 1RJ7), ICAM1 (PDB-ID: 5MZA), CASP3 (PDB-ID: 1RE1), MAPK1 (PDB-ID: 6SLG), TP53 (PDB-ID: 5ECG), PRKCA (PDB-ID: 4RA4), MAPK3 (PDB-ID: 2ZOQ), MAPK14 (PDB-ID: 6SFO), and PPARγ (PDB-ID: 8SC9), sourced from the Protein Data Bank [108]. The two-dimensional structures of compounds corresponding to the selected targets were downloaded from PubChem, and OpenBabel 2.4.1 software was used to convert these two-dimensional structures into three-dimensional configurations while minimizing their structural energy. Hydrogenation calculations for the three-dimensional structures were performed using AutoDockTools 1.5.6. The crystal structure of the target protein was retrieved from the PDB database, and PyMOL 2.6 software was utilized to dehydrate, hydrogenate, and separate the ligand. AutoDockTools was applied to define the docking grid for the active site of each target protein: X105.881_Y105.378_Z110.63 for 8IC0, X17.628_Y52.071_Z40.03 for 1RJ7, X5.196_Y12.751_Z36.215 for 5MZA, X28.042_Y102.3_Z11.394 for 1RE1, X5.223_Y0.464_Z15.757 for 6SLG, X20.535_Y21.216_Z12.092 for 5ECG, X25.195_Y1.096_Z17.157 for 4RA4, X15.003_Y7.846_Z24.552 for 2ZOQ, X6.84_Y3.852_Z15.66 for 6SFO, and X21.945_Y11.73_Z25.813 for 8SC9. Finally, molecular docking between the processed ligands, and target proteins was carried out using AutoDock Vina 1.2.0, and the results were visualized and analyzed with PyMOL 2.6 software. Hydrogen bonds play an indispensable role in the formation of complexes between small molecule ligands (active compounds) and their protein receptors. Typically, when the binding affinity value of a complex is lower than −5 kcal/mol, the interaction between the ligand and receptor is considered effective, indicating the existence of a binding relationship [109].

## 5. Conclusions

In all, the antioxidants from the bleeding sap of sponge gourd in treating TB is characterized by the combination of multi-components, multi-targets, and multi-pathways, among which the abietic acid, tretinoin, and linoleic acid might act with the PPARγ or MAPK pathway to inhibit tuberculosis. However, due to the limitation of extractions from the bleeding sap, the experiment in vitro could not be performed, so it may lack the biological validation. Additionally, in vivo experiments are further needed in the future to validate their roles during TB infection. Furthermore, future experimentation will be needed to prove that these antioxidants follow the PPARγ or MAPK pathway for tuberculosis inhibition.

## Figures and Tables

**Figure 1 ijms-26-10231-f001:**
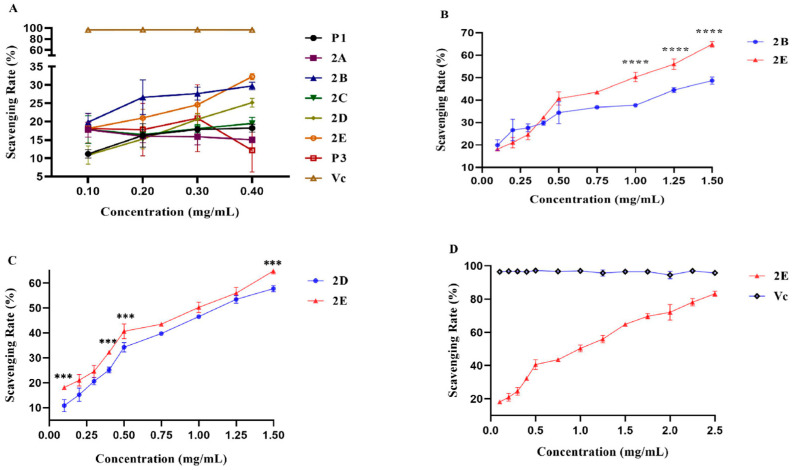
(**A**): The scavenging rate incline of different fractions; (**B**): The scavenging rate curve of fractions 2B and 2E; (**C**): The scavenging rate curve of fractions 2D and 2E; (**D**): The scavenging rate curve of fractions 2E and Vc. “****” means *p* < 0.0001, “***” means *p* < 0.01.

**Figure 2 ijms-26-10231-f002:**
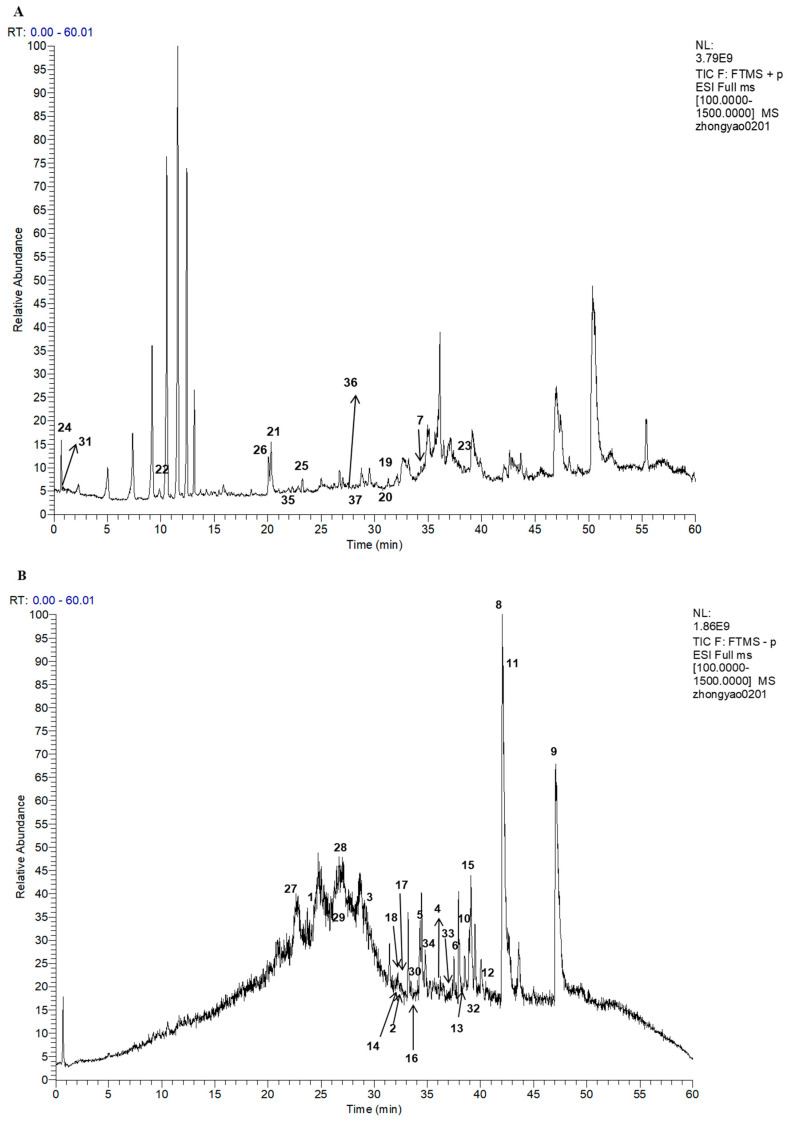
The TICs of fraction 2E in positive (**A**) and negative (**B**) modes.

**Figure 3 ijms-26-10231-f003:**
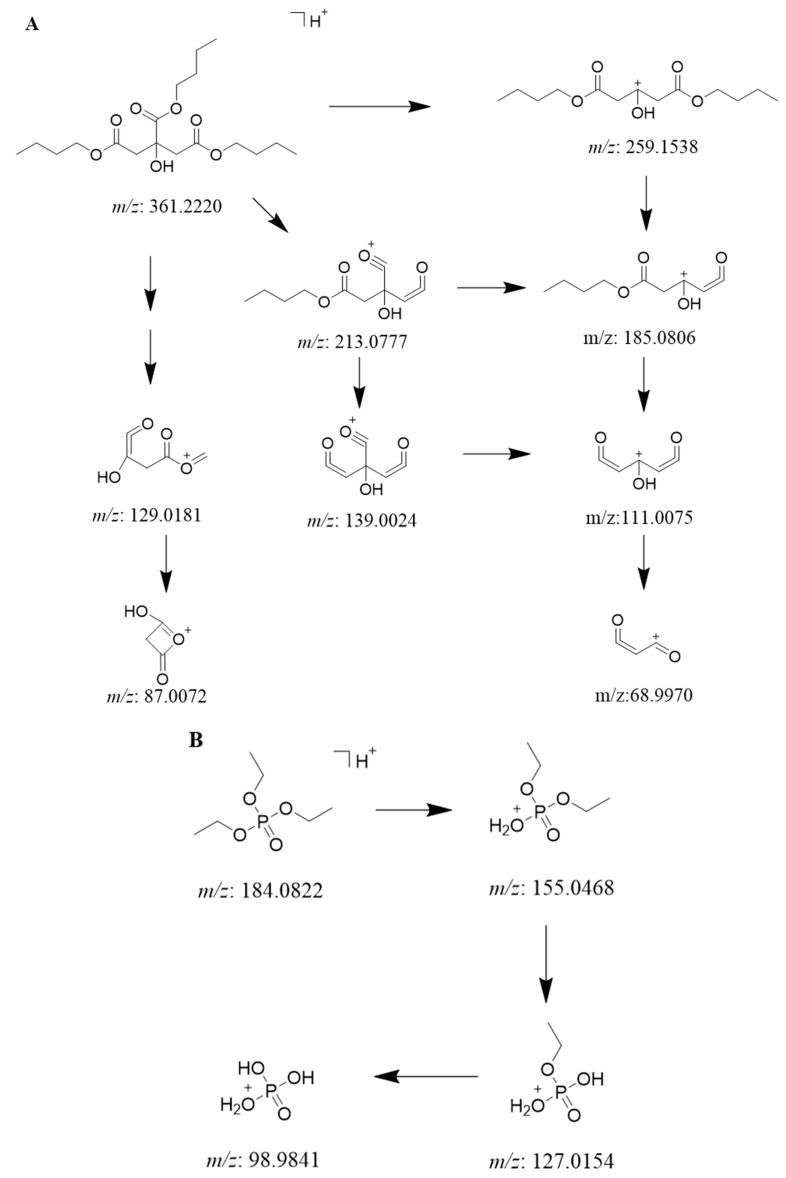
The possible fragmentation pathways of six identified compounds. (**A**): Tributyl citrate (**7**); (**B**): Triethyl phosphate (**22**); (**C**): Cinchophen (**25**); (**D**): Flavone (**35**); (**E**): Bis(2-butoxyethyl) ether (**36**); (**F**): Trenbolone (**37**). Arrow means the process of the fragmentation.

**Figure 4 ijms-26-10231-f004:**
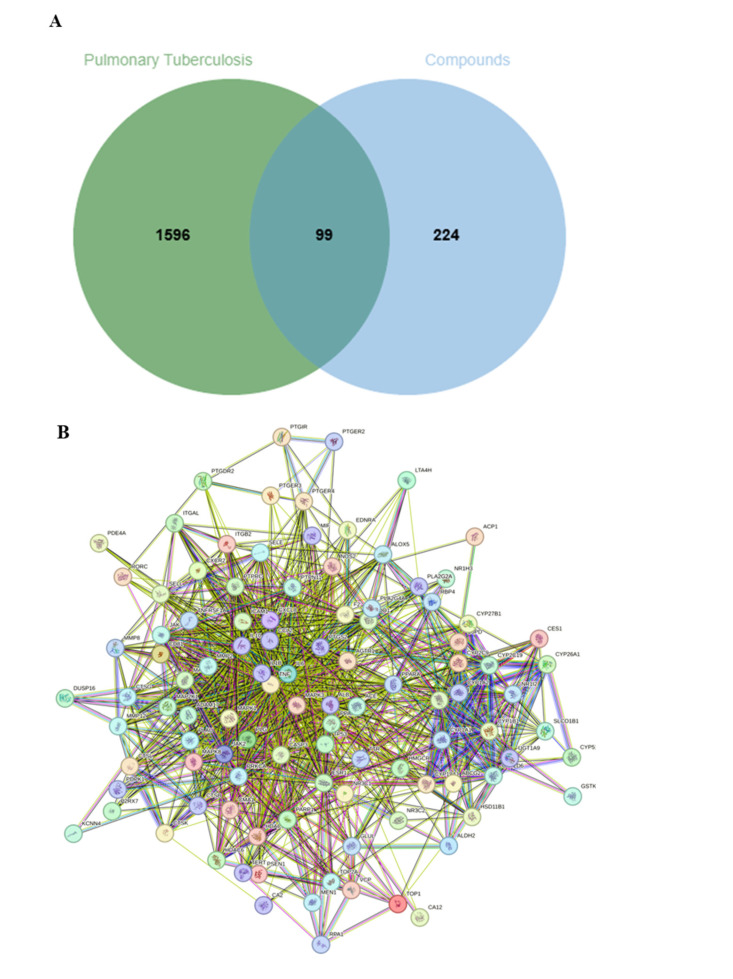
(**A**): Primary intersecting targets between TB genes and potential proteins predicted by compounds; (**B**): A PPI network diagram for primary intersecting targets; (**C**): Core targets screened from primary intersecting targets; (**D**): GO enrichment analysis of the core targets (**E**): KEGG enrichment pathways of the core targets.

**Figure 5 ijms-26-10231-f005:**
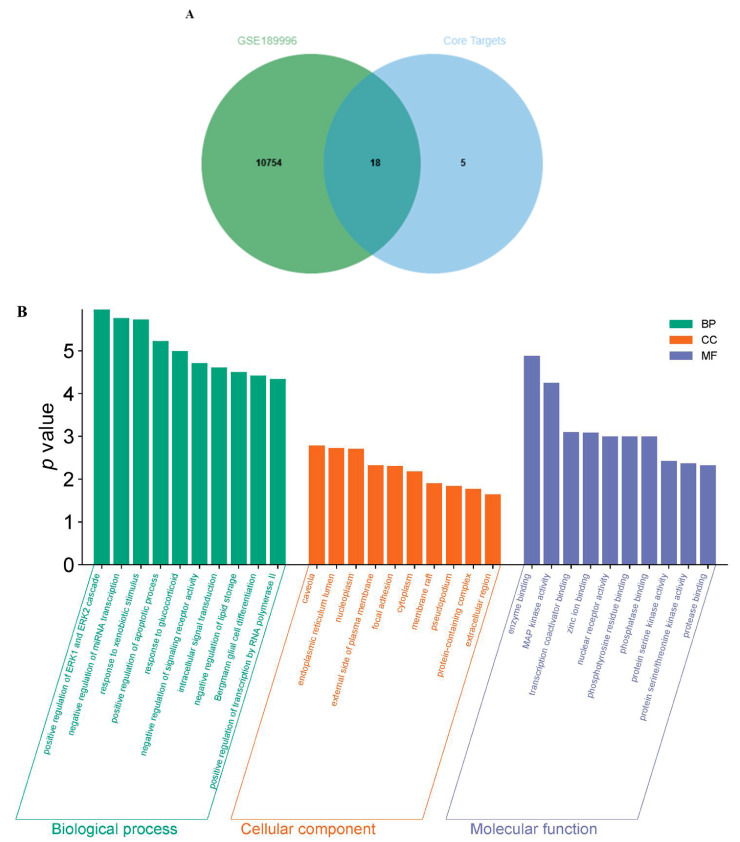
(**A**): Advanced intersecting targets between GSE189996 and core targets; (**B**): GO enrichment analysis of advanced intersecting targets; (**C**): KEGG enrichment pathways of advanced intersecting targets; (**D**): Venny map of CTP network.

**Figure 6 ijms-26-10231-f006:**
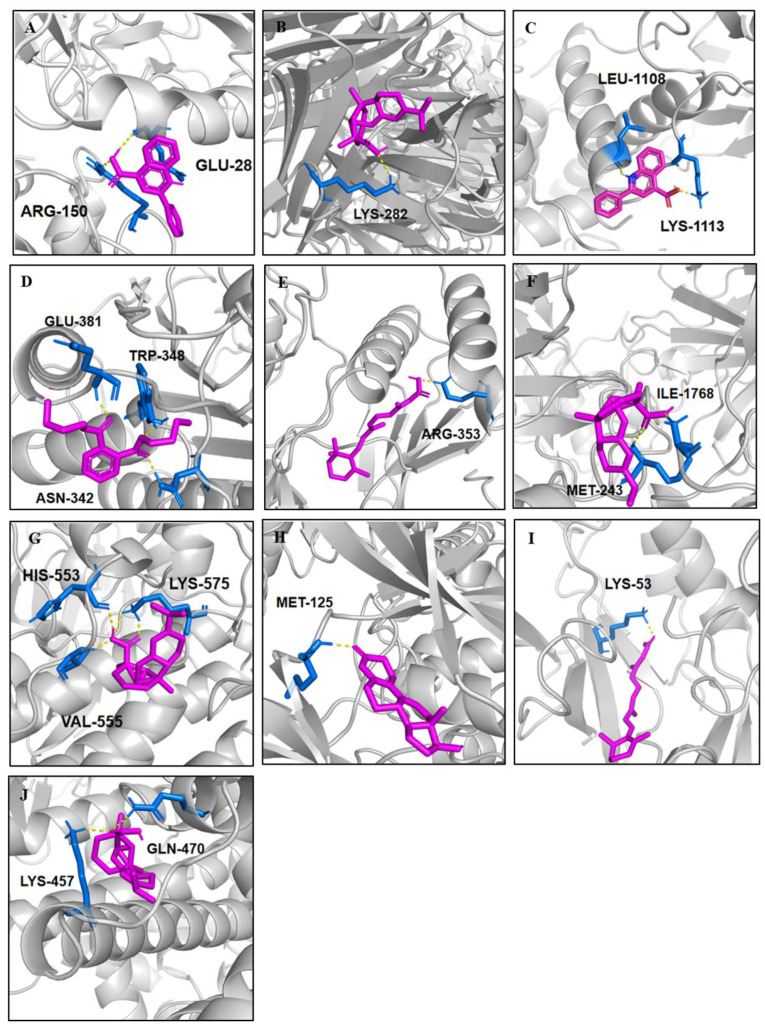
The representative molecular docking between the receptor and ligand. (**A**): CXCL8 and cinchophen (**25**); (**B**): TNF and abietic acid (**33**); (**C**): ICAM1 and cinchophen (**25**); (**D**): CASP3 and dibutyl phthalate (**31**); (**E**): MAPK1 and tretinoin (**34**); (**F**): TP53 and abietic acid (**33**); (**G**): PRKCA and abietic acid (**33**); (**H**): MAPK3 and trenbolone (**37**); (**I**): MAPK14 and tretinoin (**34**); (**J**): PPARγ and abietic acid (**33**). (The red color: the ligand molecular; the blue color: the amino acid residues in the receptor; the yellow dash: the interactions between the receptor and ligand.)

**Figure 7 ijms-26-10231-f007:**
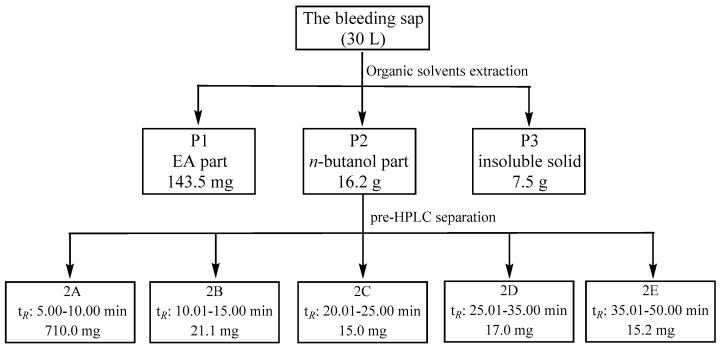
The flow chart for fractionation of bleeding sap.

**Table 1 ijms-26-10231-t001:** Identification of 37 compounds in fraction 2E by UPLC-Orbitrap-MS/MS.

No.	RT [Min]	mzCloud Best Match	Molecular Ion Peak (m/z)	Reference Ion	Name	Formula	Annot. Delta Mass [ppm]	Fragment Ions (*m*/*z*)	References
Fatty acids & esters									
**1**	24.79	99.6	157.1232	[M-H]^−1^	Nonanoic acid	C_9_H_18_O_2_	−1.09	139.1128; 113.0973; 59.0137	[40]
**2**	32.47	99.6	243.1963	[M-H]^−1^	3-Hydroxy myristic acid	C_14_H_28_O_3_	−0.95	225.1863; 181.1959; 59.0138;	[41]
**3**	28.82	99.2	171.1389	[M-H]^−1^	Decanoic acid	C_10_H_20_O_2_	−0.97	153.1285; 127.1129	[40]
**4**	36.20	99.2	213.1858	[M-H]^−1^	Tridecylic acid	C_13_H_26_O_2_	−0.99	195.1753; 168.9926	[42]
**5**	34.52	99.1	199.1700	[M-H]^−1^	Lauric acid	C_12_H_24_O_2_	−1.62	181.1600; 155.1081; 59.8251	[40]
**6**	37.48	99.1	227.2014	[M-H]^−1^	Myristic acid	C_14_H_28_O_2_	−1.31	209.1911; 59.0136	[40]
**7**	34.35	99.1	361.2220	[M+H]^+1^	Tributyl citrate	C_18_H_32_O_7_	−0.17	259.1539; 213.0778; 185.0806; 129.0181; 87.0072; 68.9970;	[43]
**8**	42.05	98.9	255.2325	[M-H]^−1^	Ethyl myristate	C_16_H_32_O_2_	−1.89	237.2226; 59.0138	[44]
**9**	47.10	98.9	283.2636	[M-H]^−1^	Stearic acid	C_18_H_36_O_2_	−2.15	265.2538	[45]
**10**	39.24	98.8	241.2170	[M-H]^−1^	Pentadecanoic acid	C_15_H_30_O_2_	−1.34	223.2068	[40]
**11**	42.72	98.6	281.2482	[M-H]^−1^	Oleic acid	C_18_H_34_O_2_	−1.43	263.2381; 127.0763; 71.0139	[40,46]
**12**	41.08	98.5	267.2327	[M-H]^−1^	trans-10-Heptadecenoic acid	C_17_H_32_O_2_	−1.07	249.2218; 83.0503	[47]
**13**	38.58	98.4	253.2169	[M-H]^−1^	Palmitoleic acid	C_16_H_30_O_2_	−1.45	235.2069; 71.0138	[48]
**14**	32.20	98.4	185.1544	[M-H]^−1^	Undecanoic acid	C_11_H_22_O_2_	−1.36	167.1441	[40]
**15**	39.62	97.7	279.2326	[M-H]^−1^	Linoleic Acid	C_18_H_32_O_2_	−1.35	261.2224; 71.0138; 59.0137	[48]
**16**	33.96	95.8	297.2433	[M-H]^−1^	(E)-6-hydroxyoctadec-4-enoic acid	C_18_H_34_O_3_	−0.88	279.2329; 183.1398; 155.1079; 141.1284; 71.0486;	[49]
**17**	33.84	93.3	293.2119	[M-H]^−1^	13-Hydroxy-9*Z*,11*E*,15*Z*-octadecatrienoic acid	C_18_H_30_O_3_	−0.96	275.2016; 185.1184; 125.0970; 97.0657	[50]
**18**	32.62	93	271.2277	[M-H]^−1^	16-Hydroxyhexadecanoic acid	C_16_H_32_O_3_	−0.65	253.2173; 225.2220; 197.1911 155.1443; 99.0815; 59.0138	[50]
Phosphorus-containing compounds									
**19**	31.25	99.7	267.1717	[M+H]^+1^	Tributyl phosphate	C_12_H_27_O_4_P	−1.15	267.1716; 211.1093; 155.0466 116.9946; 98.9839; 57.0698;	[51]
**20**	31.21	99.6	211.1092	[M+H]^+1^	Dibutyl phosphate	C_8_H_19_O_4_P	−0.93	155.0466; 98.9840; 7.0698	[51]
**21**	20.26	99.2	279.0928	[M+H]^+1^	Triphenylphosphine oxide	C_18_H_15_OP	−1.75	219.0568; 201.0462; 173.0512 141.0102; 95.0043; 77.0386;	[52]
**22**	9.96	99.1	183.0779	[M+H]^+1^	Triethyl phosphate	C_6_H_15_O_4_P	−1.13	155.0468; 127.0154; 116.9947; 98.9841	[53]
Alkaloids									
**23**	38.14	98.3	256.2634	[M+H]^+1^	Hexadecanamide	C_16_H_33_NO	−0.25	130.1225; 102.0912; 88.0756;	[54]
**24**	0.74	94.5	125.0708	[M+H]^+1^	4,6-Dimethyl-2-hydroxypyrimidine	C_6_H_8_N_2_O	−0.82	107.0603; 98.0603; 66.0338	[55]
**25**	23.27	94.1	250.0860	[M+H]^+1^	Cinchophen	C_16_H_11_NO_2_	−0.89	232.0754; 222.0909; 204.0810	[56]
**26**	20.22	93.5	225.1958	[M+H]^+1^	N, N-Dicyclohexylurea	C_13_H_24_N_2_O	−1.53	143.1179; 100.1120; 83.0855	[54]
Sulfur-containing compounds									
**27**	22.60	99.8	265.1476	[M-H]^−1^	Dodecyl sulfate	C_12_H_26_O_4_S	−1.09	96.9601; 79.9574	[57]
**28**	26.98	99.6	293.1789	[M-H]^−1^	Myristyl sulfate	C_14_H_30_O_4_S	−0.95	96.9600; 79.9573	[58]
**29**	26.52	99.3	325.1839	[M-H]^−1^	4-Dodecylbenzenesulfonic acid	C_18_H_30_O_3_S	−1.29	239.0745; 225.0588; 197.0260; 183.0119; 170.0040; 79.9574	[57,58,59]
Aromatics									
**30**	34.23	99.7	205.1595	[M-H]^−1^	2,4-Di-tert-butylphenol	C_14_H_22_O	−1.55	189.1284	[60]
**31**	0.94	99.2	279.1590	[M+H]^+1^	Dibutyl phthalate	C_16_H_22_O_4_	−0.43	205.0856; 149.0232; 121.0282 111.0438; 93.0335; 65.0386	[61,62]
**32**	38.99	99	339.2318	[M-H]^−1^	2,2′-Methylenebis(4-methyl-6-tert-butylphenol)	C_23_H_32_O_2_	−3.54	163.1127; 147.0812; 107.0503	[63]
Terpenes									
**33**	37.59	97.9	301.2170	[M-H]^−1^	Abietic acid	C_20_H_30_O_2_	−1.07	271.1695; 257.1539	[64,65,66]
**34**	35.33	96.8	299.2013	[M-H]^−1^	Tretinoin	C_20_H_28_O_2_	−1.2	265.2418; 255.2118; 175.0765	[67,68]
Flavonoid									
**35**	22.09	98.2	223.0753	[M+H]^+1^	Flavone	C_15_H_10_O_2_	−0.38	178.0776; 129.0335; 121.0282; 103.0542; 95.0488; 77.0388	[69]
Ether									
**36**	27.26	97.7	219.1952	[M+H]^+1^	Bis(2-butoxyethyl) ether	C_12_H_26_O_3_	−1.11	163.1312; 145.1215; 101.0960; 89.0598; 83.0856; 57.0698;	[70]
Steroid									
**37**	27.32	90.8	271.1691	[M+H]^+1^	Trenbolone	C_18_H_22_O_2_	−0.59	253.1588; 215.1063; 197.0961; 193.1008; 145.1012; 131.0858;	[71,72]

No. means the number of the compounds.

**Table 2 ijms-26-10231-t002:** Main molecular docking results of potential compounds and their targets on CTP network.

No.	Targeted Protein	Compounds (No. in Table 1)	Binding Energy (kcal/mol)	H-Bonds Number	Binding Site
1	CXCL8	Cinchophen (**25**)	−6.12	2	ARG-150; GLU-28
2	TNF	Abietic acid (**33**)	−6.83	1	LYS-282
3	ICAM1	Cinchophen (**25**)	−6.19	2	LEU-1108; LYS-1113
4	CASP3	Dibutyl phthalate (**31**)	−5.19	3	GLU-381; TRP-348; ASN-342
5	MAPK1	Tretinoin (**34**)	−6.67	1	ARG-353
6	TP53	Abietic acid (**33**)	−6.06	2	MET-243; ILE-1768
7	PRKCA	Dibutyl phthalate (**31**)	−5.64	1	LYS-517
		Abietic acid (**33**)	−8.13	4	HIS-553; VAL-555; LYS-575; LYS-575
8	MAPK3	Abietic acid (**33**)	−6.14	2	ARG-242; ILE-244
		Trenbolone (**37**)	−7.38	1	MET-125
9	MAPK14	Tretinoin (**34**)	−7.89	1	LYS-53
		Linoleic acid (**15**)	−5.02	2	ARG-149; ILE-147
		Dibutyl phthalate (**31**)	−5.58	1	MET-109
10	PPARγ	Oleic acid (**11**)	−5.04	2	LYS-458; LYS-457
		Linoleic acid (**15**)	−5.78	2	LYS-457; GLN-470
		Palmitoleic acid (**13**)	−5.21	1	SER-342
		Abietic acid (**33**)	−8.21	2	LYS-457; GLN-470
		Tretinoin (**34**)	−7.83	2	LEU-476; TYR-473
		Stearic acid (**9**)	−5.85	2	LYS-474; LYS-457
		trans-10-Heptadecenoic acid (**12**)	−5.87	2	ARG-288; GLU-343

## Data Availability

Data are contained within the article and Appendix A.

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
