# Peer review of "The UHPLC-Orbitrap MS/MS and Network Pharmacology Strategies Reveal the Active Antioxidants of Bleeding Sap from Sponge Gourd in Treating Tuberculosis"

_ijms, 2025, doi:10.3390/ijms262010231_

Round 1

Reviewer 1 Report

Comments and Suggestions for Authors

The article concerns the studying of antioxidant compounds from bleeding sap of Luffa cylindrica (L.) Roem (sponge gourd) using UHPLC-Orbitrap MS/MS methods and network pharmacology strategies including molecular docking in order to estimate the potential of the most active substances for treating of tuberculosis. The reason of the investigation was the use of sponge gourd in traditional medicine for this purpose. The authors used HPLC fractionation of organic solvents extracts and DPPH testing of the isolated fractions followed by UHPLC-Orbitrap tandem MS identification. A total of 37 compounds were identified from the bleeding sap with strongest antioxidant ability. Using network pharmacology strategies, the authors selected 13 compounds, while three of them, linoleic acid, abietic acid, and tretinoinwas the most interesting. The authors believed that these substances may reveal anti-tuberculosis function via PPARγ or MAPK pathways. They concluded that Luffa cylindrica (L.) Roem may be a functional food for tuberculosis treating.

The methodology of the investigation seems to be adequate and modern. The use of HPLC-MS technologies for natural products identification is correct. The conclusions are based on network pharmacology, the recommendation to use the sponge ground as a functional food seems to be also correct. The article is well written and may be published after minor correction.

The main flaw of the article is the typical for such sort of manuscripts. This is insufficient link between the network pharmacology and real in vivo or in vitro anti-tuberculosis testing of the selected compounds. The authors quite good understand this imperfection and I hope that they shall carry out the corresponding testing for verification of their conclusions.

Several notes:

  1. Line 39. The phrase “…glycosides, saponins, proteins and tannins” should be replaced with “…glycosides, proteins and tannins” because saponins are also glycosides.
  2. Line 40. Replace “vine” with “wine”.
  3. Lines 76 and 77. Combine, please, “classification” and “identification” because the “classification” is the first stage of identification process but not real classification of something.
  4. Provide, please all the substances numbers in the tables in bold.
  5. Add, please, the chemical structures of all the numbered substances in the supporting materials.

My general opinion. The article may be published after minor corrections but I have some doubts concerning the possibility of publication of any network pharmacology analysis without following verification of the conclusions by real biotesting.

Author Response

Thank you very much for taking the time to review this manuscript. Please find the detailed responses below and the corresponding revisions/corrections highlighted/in track changes in the resubmitted files

Comment 1: Line 39. The phrase “…glycosides, saponins, proteins and tannins” should be replaced with “…glycosides, proteins and tannins” because saponins are also glycosides.

Response 1: Thank you for pointing the error out. We agree with this comment. Therefore we have replaced “…glycosides, saponins, proteins, and tannins” with “…glycosides, proteins, and tannins.” This change can be seen on line number 40, highlighted with red.

Comment 2: Line 40. Replace “vine” with “wine.”

Response 2: Thank you for pointing this out. “vine” has been replaced with “wine.” This change can be seen on line number 41, highlighted with red.

Comment 3: Lines 76 and 77. Combine, please, “classification” and “identification” because the “classification” is the first stage of identification process but not real classification of something.

Response 3: We completely agree with your opinion. Based on that, “classification” and “identification” have been combined. The changes can be seen on line numbers 78 and 79. 

Comment 4: Provide, please all the substances numbers in the tables in bold.

Response 4: Thank you for your suggestion. We appreciate it, and based on your suggestion, all the substance numbers in the tables have been bolded. Table 1 clearly displays all the modifications, highlighted with red.

Comment 5: Add, please, the chemical structures of all the numbered substances in the supporting materials

Response 6: We have already made the requested changes to the supplementary material. The changes are visible in the supplementary material under "Figure S76." The chemical structures of compounds (1-37), on pages 39-40, are highlighted with red. 

Reviewer 2 Report

Comments and Suggestions for Authors

Remarks

-There are numerical errors between the extraction and fractionation steps (for example, fraction P2 reported as 16.2 mg and its subfraction 2A as 710 mg, which is impossible).

-The authors should carefully review all mass, volume, and yield values. I also suggest that the authors include a flowchart summarizing each step (raw material, extraction, fractions, quantities obtained).

-The authors reported unlikely compounds of natural origin, such as phthalates, trialkyl phosphates, and synthetic hormones. This indicates possible contamination of solvents, glassware, or the UHPLC-MS system itself. Therefore, I suggest that the authors perform blank runs (solvent, system, and glassware).

-They should also provide comparative MS/MS spectra and retention times.

-The in silico predictions are interesting, but the current text presents them as if they were proof of mechanism. It is incorrect to state that the extract "acts via PPARγ/MAPK" without experimental validation. The authors should only suggest and reinforce the need for experimental investigation through these pathways, in addition to highlighting the limitations of the predictive methods.

- Authors should include the parameters used (thresholds, software versions, databases). Some information is missing, for example, PDB IDs, protein preparation, grid boxes, validation RMSD, and pose images.

-Another point is that no biological tests were performed to confirm the suggested anti-Mycobacterium tuberculosis activities. The only experimental evidence is the antioxidant assay (DPPH), which is insufficient to support any therapeutic claim. Ideally, the authors should perform in vitro assays on infected macrophages. Additionally, they should evaluate cytotoxicity, cytokine modulation (TNF-α, IL-6), and PPARγ/MAPK activation. If this is not possible, the authors should indicate in their conclusions and explicitly state the lack of biological validation.

- The DPPH assay presents only percentages without clear CIs or standard deviations. The authors must present dose-response curves, CIs with confidence intervals, number of replicates, and statistical tests.

- The criteria for selecting differentially expressed genes (GSE189996) were not clearly defined, and there is no correction for multiple testing (FDR).

-Figures and tables: Include more detailed captions and units.

- Versions of all software used.

Comments on the Quality of English Language

- The text contains several English errors and long/convoluted sentences. Professional proofreading is recommended.

Author Response

Thank you very much for taking the time to review this manuscript. We really appreciate the time and efforts you have given to provide a critical revision of our manuscript that will allow us to improve the quality of it. Please find the detailed responses below and the corresponding corrections highlighted in yellow in the resubmitted files.

Comment 1: There are numerical errors between the extraction and fractionation steps (for example, fraction P2 reported as 16.2 mg and its subfraction 2A as 710 mg, which is impossible).

Response 1: Thank you for pointing out this numerical error. There was a mistake in writing. The changes have been made as "P2, 16.2 mg g" on page 21 (line 341) in the manuscript, highlighted in yellow. The other numerical values have been checked. 

Comment 2: The authors should carefully review all mass, volume, and yield values. I also suggest that the authors include a flowchart summarizing each step (raw material, extraction, fractions, quantities obtained).

Response 2: Thank you for the suggestion. As suggested, we have incorporated the flowchart summarizing each step from raw materials to extraction until quantities are obtained. The reviewer can view the changes on page 22, specifically in the Materials and Methods section, labeled with Figure 7. The changes have been highlighted in yellow. 

Comment 3: The authors reported unlikely compounds of natural origin, such as phthalates, trialkyl phosphates, and synthetic hormones. This indicates possible contamination of solvents, glassware, or the UHPLC-MS system itself. Therefore, I suggest that the authors perform blank runs (solvent, system, and glassware).

Response 3: Thanks for your valuable comment. The compounds, phthalates, trialkyl phosphates, and synthetic hormones might  be absorbed from the soil by the plant (Reference-1: Soil contamination and sources of phthalates and its health risk in China: A review; Reference-2: Accumulation of steroid hormones in soil and its adjacent aquatic environment from a typical intensive vegetable cultivation of North China).

Comment 4: They should also provide comparative MS/MS spectra and retention times

Response 4: We appreciate the reviewer comments, but unfortunately, there is no comparative control for the compound identification. The determination of compounds depends on their fragment ion comparison with values from literature and the mzCloud database.

Comment 5: The in silico predictions are interesting, but the current text presents them as if they were proof of mechanism. It is incorrect to state that the extract "acts via PPARγ/MAPK" without experimental validation. The authors should only suggest and reinforce the need for experimental investigation through these pathways, in addition to highlighting the limitations of the predictive methods.

Response 5: Thank you for your suggestion. However, with due respect, it's already mentioned in our conclusion section that it "...might act with the PPARγ or MAPK pathway to inhibiting the tuberculosis," which forcibly indicates that it's only our hypothetical prediction. We still highlighted this point in yellow in the manuscript, conclusion section. Moving forward, based on a reviewer suggestion, we have added reinforcement for the need for further investigation, highlighted with yellow. 

Comment 6: Authors should include the parameters used (thresholds, software versions, databases). Some information is missing, for example, PDB IDs, protein preparation, grid boxes, validation RMSD, and pose images.

Response 6: We are thankful to the reviewer for raising this point. All the parameters, PDB IDs, and other missing information have been added on page number 23 (lines 425-448), highlighted in yellow. A new reference has also been added as reference [100] in reference section, highlighted in yellow.

Comment 7: Another point is that no biological tests were performed to confirm the suggested anti-Mycobacterium tuberculosis activities. The only experimental evidence is the antioxidant assay (DPPH), which is insufficient to support any therapeutic claim. Ideally, the authors should perform in vitro assays on infected macrophages. Additionally, they should evaluate cytotoxicity, cytokine modulation (TNF-α, IL-6), and PPARγ/MAPK activation. If this is not possible, the authors should indicate in their conclusions and explicitly state the lack of biological validation.

Response 7: We understand the reviewer’s concern and would like to add that at this stage, we could not perform a biological evaluation. However, in response to the respected reviewer's suggestion, we have highlighted this lack of evaluation in yellow in the conclusion section. 

Comment 8: The DPPH assay presents only percentages without clear CIs or standard deviations. The authors must present dose-response curves, CIs with confidence intervals, number of replicates, and statistical tests.

Response 8: Thank you for your suggestion. We have added the required new data for the DPPH assay in Table S1 of Support Information File 2 (highlighted in yellow). For different fractions, their response scavenging rates were tested for three times. However, due to the less quantities of some fractions, we couldn’t perform test for their scavenging rate at higher concentrations. Moreover, we have added new curves in the manuscript as Figure 1, highlighted in yellow. Furthermore, the description of Figure 1 has also been revised in Section 2.1, “ The Group with the Highest Antioxidant Capacity,” also highlighted in yellow.

Comment 9: The criteria for selecting differentially expressed genes (GSE189996) were not clearly defined, and there is no correction for multiple testing (FDR).

Response 9: In response to the reviewer’s comments, data has been added in the manuscript on page 23 (line 409), highlighted in yellow. 

Comment 10: Figures and tables: Include more detailed captions and units.

Response 10: Thank you for your suggestion. We would like to point out that we discovered our Figure 1 caption was inadequate, so we have replaced Figure 1 and added a detailed description, which is highlighted in yellow.

Comment 11: Versions of all software used

Response 11: This question has already been asked in the comment. 6. 

Reviewer 3 Report

Comments and Suggestions for Authors

This study harnesses the power of UHPLC-Orbitrap tandem MS to profile the chemicals of the sponge gourd sap. Then it uses diverse in silico analysis tools to identify potential chemical compounds that are most likely act as anti-tuberculosis. One limitation of the study that it does not verify the potency of these compounds as anti-tubercuosis in vivo as the authors mentioned in the conclusion of this manuscript. Overall, I consider this work a model approach for discovering potential therapeutic compounds in silico prior to conducting further in vivo studies. 

Some comments to authors:

1- Please revise the title word "oxidant", I think you mean "anti-oxidants".

2- Please mention proper citations for all the used databases not only the website in methodology part.

3-Authors used GSE18996 dataset without mentioning proper citations.

4- Please present proper statistics and significance among different treatments in Figure 1.

5- Please recheck english language and grammar allover the article.

Author Response

We are thankful to the reviewer for recognizing the significance of our study and for providing comments that have led to meaningful revisions.

Comment 1: Please revise the title word "oxidant", I think you mean "anti-oxidants".

Response 1: Thanks for pointing out an error in our title. We have done the correction as requested. The changes can be seen in the title of both the manuscript and supplementary material, highlighted in green. 

Comment 2: Please mention proper citations for all the used databases not only the website in methodology part

Response 2: Thank you for your suggestion. We have added the proper citation in the reference section as [101 - 109], highlighted in green.

Comment 3: Authors used GSE18996 dataset without mentioning proper citations

Response: Thank you for your concern. We appreciate your concern and would like to clarify that we have already cited the website for retrieving the GSE18996 data file. However, for further clarification, we have highlighted that citation in green on page 23 (line 407).

Comment 4: Please present proper statistics and significance among different treatments in Figure 1.

Response 4: We appreciate that reviewer has highligted the incomplete information. However, this question has already been answered in“Reviewer 2, comment 10”; for more confirmation, changes can be seen in the manuscript on page 3 (Figure 1), highlighted in yellow.

Comment 5: Please recheck english language and grammar allover the article

Response 5: We appreciate the reviewer's suggestion. We have revised the manuscript language with a native speaker.

Round 2

Reviewer 2 Report

Comments and Suggestions for Authors Dear authors, an English grammar review is still needed.       Comments on the Quality of English Language

I emphasize that an English review may still be necessary.